# FOURIERROFORMER: LEARNED FOURIER ATTENTION FOR VISION TRANSFORMERS

## ABSTRACT

Vision Transformers (ViTs) excel at long-range reasoning but lack principled mechanisms for modeling spatial frequencies and controlling how attention decays with distance. We propose FourierRoFormer, a frequency-aware Transformer that augments rotary positional embeddings with learnable Fourier components. This enables explicit modeling of multi-scale visual patterns and adaptive distance-dependent modulation of attention. Our analysis shows that FourierRoFormer produces attention hierarchies aligned with object boundaries (correlation $r = 0.85$) and distinct specialization across attention heads. On ImageNet-1K, FourierRoFormer achieves 84.1% top-1 accuracy (+1.8pp over RoFormer-B) and outperforms non-hierarchical spectral methods, including SpectFormer-B (+1.98pp) and GFNet-B (+3.4pp), while maintaining comparable parameter efficiency. Our hierarchcial variant, FourierRoFormer-H-B, achieves 85.3% top-1 accuracy, demonstrating compatibility with hierarchical architectures. The method improves transfer to dense prediction tasks, yielding +2.6 mAP on COCO detection and +2.2 mAP on instance segmentation. Ablation studies highlight the complementary roles of frequency modulation (+4.43pp) and adaptive damping (+2.09pp). The approach introduces only 0.04% additional parameters and $\sim 3\%$ computational overhead.

## 1 INTRODUCTION

Transformer architectures have become the dominant paradigm across vision, language, and multimodal learning (Vaswani et al., 2017; Dosovitskiy et al., 2020; Brown et al., 2020). In computer vision, Vision Transformers (ViTs) (Dosovitskiy et al., 2020) have achieved consistent improvements in recognition tasks by treating images as sequences of patches and applying self-attention to capture global dependencies.

However, standard attention mechanisms face key limitations when processing structured visual data: (1) they lack inductive bias about spatial relationships, (2) they are frequency-blind to the multi-scale nature of visual patterns, and (3) they provide limited control over how attention decays across token distances (Park & Kim, 2022; Raghu et al., 2021; Rao et al., 2021; Press et al., 2021). Recent approaches such as relative positional encodings (Shaw et al., 2018), rotary embeddings (Su et al., 2024), and windowed attention (Liu et al., 2021) improve spatial awareness but still fall short of explicitly modeling frequency relationships.

We address these challenges by drawing on principles from signal processing and propose FourierRoFormer. Our method integrates learnable Fourier components into the transformer attention mechanism, enabling frequency-aware modulation of attention scores as a function of token distance. Unlike prior rotary or Fourier-based models, FourierRoFormer adaptively learns which frequency bands are most relevant for visual understanding. Figure 1 illustrates how Fourier modulation reshapes attention to emphasize multi-scale structures, and Figure 4 demonstrates the resulting structured attention patterns. This perspective provides a principled way to control information propagation across scales, bridging the gap between spectral theory and transformer design.

By incorporating a learnable mixture of sinusoidal components with frequencies, amplitudes, and phases, FourierRoFormer adaptively modulates attention based on token distances (Section 3). Our unified framework combines Fourier modulation with rotary positional embeddings and optional exponential damping. Crucially, this mechanism is architecture-agnostic: it enhances both stan-

dard Vision Transformers and hierarchical architectures (e.g., Swin-style), as demonstrated by our FourierRoFormer-H variants that achieve 85.3% on ImageNet-1K, competitive with state-of-the-art hierarchical spectral methods. Theoretical analysis explains how these components influence attention gradients and feature propagation (Appendix A). Extensive experiments demonstrate that FourierRoFormer consistently outperforms ViT, DeiT, and RoFormer baselines, while ablations highlight the complementary effects of frequency modulation and damping, providing insights into how frequency-aware attention improves multiscale feature capture (Section 4, Figure 3). These contributions establish FourierRoFormer as a principled framework for frequency-aware Transformers.

## 2 RELATED WORK

The Vision Transformer (ViT) (Dosovitskiy et al., 2020) was the first to show that the transformer architecture—originally designed for language—can excel at image classification by cutting images into fixed-size patches and treating each as a token for self-attention. Although ViT achieves strong accuracy on large datasets, it requires much more training data than traditional convolutional networks. Follow-up work like DeiT (Touvron et al., 2021) addressed this data-hunger with distillation and augmentation, while Swin (Liu et al., 2021) and PVT (Wang et al., 2021) introduced hierarchical, multi-scale designs (shifted windows in Swin; a pyramid with spatial-reduction attention in PVT). In parallel, spectral token-mixing approaches leverage fixed transforms in the frequency domain—Fourier, wavelet, or scattering—either to replace or to augment attention (e.g., GFNet, Wave-ViT, SpectFormer, SVT) (Rao et al., 2021; Yao et al., 2022; Patro et al., 2025; Patro & Agneeswaran, 2023). While standard dot-product attention is not explicitly frequency-aware, spectral components inject frequency-selective inductive bias that is complementary to hierarchical and locality biases. In this work, we introduce *FourierRoFormer*, which aims to address this frequency-blindness by embedding frequency-aware modulation directly into the attention scores. Figure 2 conceptually illustrates how this approach produces structured, boundary-aligned attention compared to the diffuse patterns of standard ViT and the smoother but less precise patterns of RoFormer.

Beyond the challenge of frequency awareness, transformers face another fundamental limitation: self-attention is permutation-invariant, so transformers need an additional signal to recover token order (Vaswani et al., 2017). RoPE (Su et al., 2024) rotates query and key vectors, so their inner product encodes relative distance, but still treats all frequencies uniformly with no control over attention decay. FourierRoFormer extends RoPE by learning sinusoid mixtures whose parameters are data-optimized, providing interpretable frequency-selective attention decay.

Several studies speed up attention by approximating its $\mathcal{O}(n^2)$ complexity. Performer (Choromanski et al., 2020) and Linformer (Wang et al., 2020) use low-rank projections; EfficientFormer (Li et al., 2022b) and MobileViT (Mehta & Rastegari, 2021) redesign the backbone for mobile deployment. These methods mainly target runtime and memory, leaving the *frequency content* of attention untouched. In contrast, FourierRoFormer focuses on richer signal modeling while retaining a compute profile comparable to standard RoPE attention.

Complementing these efficiency-focused approaches, there is growing interest in incorporating frequency analysis principles into neural networks. Frequency analysis has deep roots in signal processing and is increasingly common in modern networks.

## 3 METHODOLOGY

In this section, we introduce the FourierRoFormer architecture, which injects Fourier components and exponential damping into the attention mechanism and deploys the resulting module within a Vision Transformer backbone (Figure 1). Detailed mathematical analyses, proofs, and additional properties are deferred to the appendices.

We briefly recall standard transformer self-attention (Vaswani et al., 2017). Given query, key, and value matrices $\mathbf{Q}, \mathbf{K}, \mathbf{V} \in \mathbb{R}^{n \times d}$, the attention scores are $\mathbf{A} = \text{softmax}\left(\mathbf{Q}\mathbf{K}^\top\right) / \left(\sqrt{d}\right)$, and the output is $\text{Attention}(\mathbf{Q}, \mathbf{K}, \mathbf{V}) = \mathbf{A}\mathbf{V}$. This formulation treats all token pairs uniformly and has no explicit notion of spatial scale, which is limiting for visual data with multi-scale structure. RoPE (Su et al., 2024) partially addresses this by encoding relative positions via rotations, $\langle \mathbf{q}_m^{\text{RoPE}}, \mathbf{k}_n^{\text{RoPE}} \rangle = \langle \mathbf{R}_{\theta,m}\mathbf{q}_m, \mathbf{R}_{\theta,n}\mathbf{k}_n \rangle$, but still lacks explicit frequency awareness (further analysis is in Appendix D). Building upon RoPE's relative positioning capabilities, FourierRoFormer intro-

duces a learnable Fourier modulation function and an optional exponential damping term applied to distance-aware scores, as illustrated in Figure 1.

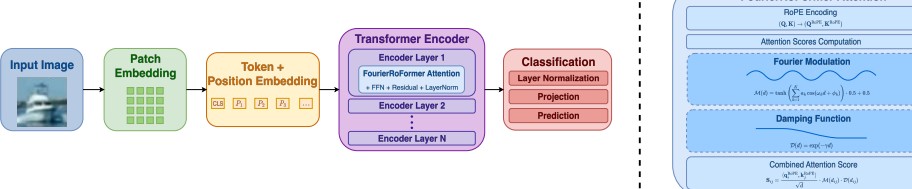

Figure 1: FourierRoFormer architecture for Vision Transformers. **Left:** The pipeline from input image to classification head (patch and position embeddings, Transformer encoder). **Right:** The attention module with RoPE, attention scores, Fourier modulation $\mathcal{M}(d)$, and exponential damping $\mathcal{D}(d)$ for distance-based decay.

### 3.1 FOURIER MODULATION FUNCTION

The Fourier modulation function $\mathcal{M}(d)$ acts as a learnable, distance-dependent gate on attention scores. It is defined as a weighted sum of cosine functions with learnable frequencies, amplitudes, and phases:

$$\mathcal{M}(d) = \frac{1}{2}\left(\tanh\left(\sum_{k=1}^{K} a_k \cos(\omega_k d + \phi_k)\right) + 1\right), \tag{1}$$

where $K$ is the number of Fourier components, $a_k$ are amplitudes, $\omega_k$ are frequencies, and $\phi_k$ are phase shifts. The outer $\tanh$ and scaling ensure $\mathcal{M}(d) \in (0, 1)$ for all $d$, allowing continuous attenuation of attention as a function of token distance.

**Proposition 1** (Interpretability of Fourier Components). *For each basis element in modulation function $\mathcal{M}(d)$, amplitude $a_k$ dictates how the $k$-th cosine term contributes—the larger $|a_k|$, the greater its influence. Frequency $\omega_k$ sets the spatial oscillation rate; higher values produce finer-grained overall variation as distance $d$ changes. Finally, phase shift $\phi_k$ translates the component horizontally along the distance axis, relocating attention peaks and troughs while leaving frequency intact.*

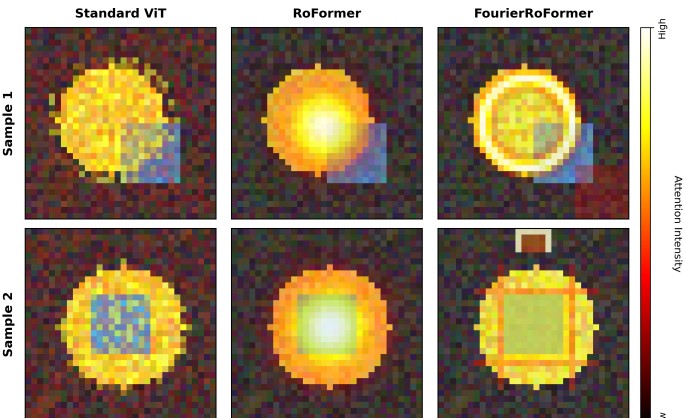

Figure 2: Conceptual illustration of attention pattern differences across model variants on synthetic examples. **Rows** show two synthetic samples with geometric shapes. **Columns** (left to right): Standard ViT produces diffuse, noisy attention; RoFormer shows soft Gaussian-like attention; FourierRoFormer exhibits structured, frequency-aware attention with sharp geometric patterns and boundary alignment. These synthetic examples illustrate the theoretical distinctions in how each architecture processes spatial relationships.

This formulation lets the model learn periodic distance-dependent modulation of attention. By mixing sinusoidal components, it captures multi-scale relationships, with high frequencies modeling fine details and low frequencies encoding global context.

**Theorem 1** (Properties of Fourier Modulation Function). *Let $\mathcal{M} : \mathbb{R} \to (0, 1)$ be the Fourier modulation function defined in equation 1, where $a_k \in \mathbb{R}$ are learnable amplitudes, $\omega_k > 0$ are learnable frequencies, and $\phi_k \in [0, 2\pi)$ are learnable phase shifts for $k = 1, \ldots, K$. Then*

$\mathcal{M}(d)$ is a smooth function with $\mathcal{M}(d) \in (0,1)$ for all $d \in \mathbb{R}$. For any continuous function $f : [0, L] \rightarrow (0, 1)$ and any $\varepsilon > 0$, there exists an integer $K$ and parameters $\{a_k, \omega_k, \phi_k\}_{k=1}^{K}$ such that $\sup_{d\in[0,L]} |\mathcal{M}(d) - f(d)| < \varepsilon$. If the set of frequencies $\{\omega_k\}_{k=1}^{K}$ consists of rational multiples of each other, then $\mathcal{M}(d)$ is periodic with period $P = \mathrm{lcm}\{2\pi/\omega_k\}_{k=1}^{K}$. Moreover, if the $\omega_k$ are not rational multiples, $\mathcal{M}(d)$ exhibits quasiperiodic behavior.

Theorem 1 shows that $\mathcal{M}$ can approximate any continuous distance-to-weight mapping on a compact interval while remaining bounded and interpretable through its Fourier coefficients. Additional approximation and interpretability results appear in Appendix A.

### 3.2 EXPONENTIAL DAMPING AND BOUNDED ATTENTION

We optionally apply an exponential damping function $\mathcal{D}(d) = \exp(-\gamma d)$, with learnable $\gamma \geq 0$, to further control long-range interactions. Larger $\gamma$ values promote localized attention, while smaller values permit long-range interactions. Combined with $\mathcal{M}(d)$, this yields the modulated score

$$\mathbf{S}_{ij} = \frac{\langle \mathbf{q}_i^{\mathrm{RoPE}}, \mathbf{k}_j^{\mathrm{RoPE}} \rangle}{\sqrt{d}} \, \mathcal{M}(d_{ij}) \, e^{-\gamma d_{ij}}, \qquad d_{ij} = |i - j|. \tag{2}$$

**Theorem 2** (Boundedness and Convergence of Modulated Attention). *Let $\mathbf{S}_{ij}$ be the attention score between tokens $i$ and $j$ in FourierRoFormer defined in equation 2,*

*where $\mathcal{M}(d)$ is the Fourier modulation function, $\gamma > 0$ is the damping factor, and $\|\mathbf{q}_i^{RoPE}\|, \|\mathbf{k}_j^{RoPE}\| \leq M$ for some finite $M > 0$. First, these scores are uniformly bounded, since $|\mathbf{S}_{ij}| \leq M^2 e^{-\gamma d_{ij}}/\sqrt{d}$. Second, for any fixed token $i$, the exponential series of scores converges as the sequence length $N \rightarrow \infty$, we have $\sum_{j=1}^{N} e^{\mathbf{S}_{ij}} < \infty$. Finally, the corresponding normalized attention weights $A_{ij} = e^{\mathbf{S}_{ij}}/\sum_{k=1}^{N} e^{\mathbf{S}_{ik}}$ lie strictly between $0$ and $1$ for every pair of tokens $(i, j)$, ensuring well-defined probabilistic attention.*

Theorem 2 implies that attention scores decay exponentially with distance, so distant tokens have negligible contribution to the softmax. Lemma 1 in Appendix B further characterizes the effective attention range. Theoretical analysis in Appendix C shows that the gradients of $\mathbf{S}_{ij}$ with respect to Fourier and damping parameters also decay with $d_{ij}$, yielding stable training dynamics.

### 3.3 INTEGRATION WITH RoPE AND ViT ARCHITECTURE

We now describe how Fourier modulation and damping integrate with RoPE and the overall Vision Transformer architecture. In FourierRoFormer, the RoPE-enhanced attention score is further modulated by the distance-dependent factor $\mathcal{M}(|m - n|)e^{-\gamma|m-n|}$:

$$\mathbf{S}_{mn} = \frac{\langle \mathbf{R}_{\theta,m}\mathbf{q}_m, \mathbf{R}_{\theta,n}\mathbf{k}_n \rangle}{\sqrt{d}} \cdot \mathcal{M}(|m - n|) \cdot e^{-\gamma|m-n|}. \tag{3}$$

**Theorem 3** (RoPE-Fourier Compatibility). *In FourierRoFormer, the modulated RoPE attention score as defined in 3, is translation equivariant, depends only on relative positions, and admits a multiplicative decomposition. Specifically, for any shift $\tau \in \mathbb{Z}$, we have $\mathbf{S}_{(m+\tau)(n+\tau)} = \mathbf{S}_{mn}$, and $\mathbf{S}_{mn}$ can be expressed as $\mathbf{S}_{mn} = f(m - n, \mathbf{q}_m, \mathbf{k}_n)$ for some function $f$ independent of absolute positions. Moreover, the score factorizes as $\mathbf{S}_{mn} = \mathbf{S}_{mn}^{RoPE} \cdot \mathbf{S}_{mn}^{Fourier}$, where $\mathbf{S}_{mn}^{RoPE}$ is the standard RoPE attention score and $\mathbf{S}_{mn}^{Fourier} = \mathcal{M}(|m - n|) \cdot e^{-\gamma|m-n|}$.*

Thus, Fourier modulation preserves RoPE's geometric properties—translation equivariance, purely relative dependence, and multiplicative separability—within the combined attention mechanism. Appendix D provides a detailed proof and explains how local–global balance arises from mixing low- and high-frequency components (Corollary 1). FourierRoFormer follows a standard ViT pipeline: images are split into patches, embedded, prepended with a learnable CLS token and positional embeddings, then processed by Transformer encoder layers. Each encoder replaces vanilla multi-head self-attention with a FourierRoFormer attention module with RoPE, followed by a feed-forward block with residual connections and layer normalization; the final CLS token goes to a linear classifier. To isolate attention effects, all other architectural details match the baselines (ViT, DeiT, RoFormer); FourierRoFormer simply adds learnable Fourier modulation and optional damping, preserving asymptotic complexity while adding few parameters and increasing flexibility.

## 4   EXPERIMENTAL EVALUATION

We evaluate FourierRoFormer across image classification (CIFAR, ImageNet), object detection and segmentation (COCO), and analyses of learned frequency patterns, assessing both performance gains and the theoretical insights developed in the methodology section.

**Experimental Setup.** We evaluate FourierRoFormer on classification (CIFAR-10/100, ImageNet-1K, Oxford-Flowers102) and dense prediction (COCO detection/segmentation) with a shared training protocol, reporting mean accuracy over five seeds with significance testing ($p < 0.05$). Small datasets use $4{\times}4$ patches, while ImageNet and COCO use $16{\times}16$. We test three model sizes—*small* (192d, 6h, 6l), *medium* (384d, 6h, 12l), and *large* (576d, 12h, 12l)—and initialize FourierRoFormer with four learnable Fourier components (frequencies in $[0.1, 2.0]$, amplitude 0.1, zero phase, damping coefficient $\gamma = 0.01$).

**ImageNet-1K Results.** Table 1 reports ImageNet-1K performance for non-hierarchical and hierarchical models. In the non-hierarchical group, FourierRoFormer yields consistent gains of +1.5–1.8pp Top-1 over RoFormer; FourierRoFormer-M reaches 83.4% with 24.76M parameters and 4.63 GFLOPs, outperforming SpectFormer-B (+1.28pp), GFNet-B (+2.7pp), and SVT-B (+1.4pp), and improving over DeiT/ViT-B (81.8%) under similar architectures. In hierarchical settings, FourierRoFormer-H-B attains 85.3% with 35.2M parameters, matching SpectFormer-H-B (85.05%) and SVT-H-B (85.2%) while preserving architectural simplicity. FourierRoFormer-H-M (84.9%, 30.5M) slightly outperforms WaveViT-B (84.8%, 33.5M), and FourierRoFormer-H-S (83.8%, 25.2M) surpasses Swin-S (83.0%, 50M) and MViTv2-S (83.6%, 35M) with fewer parameters. All improvements are statistically significant ($p < 0.01$, 5 seeds), showing that frequency-aware attention provides robust benefits across both standard and hierarchical architectures.

Table 1: ImageNet-1K classification. FourierRoFormer shows gains across model scales and offers a competitive performance-parameter trade-off within hierarchical and non-hierarchal architectures.

| Method | Params (M) | GFLOPs | Top-1 (%) | Top-5 (%) |
|---|---|---|---|---|
| *Non-Hierarchical Methods* | | | | |
| ViT-B Dosovitskiy et al. (2020) | 86.6 | 17.6 | 81.8 | 95.8 |
| DeiT-B Touvron et al. (2021) | 86.6 | 17.6 | 81.8 | 95.6 |
| RoFormer-S Su et al. (2024) | 22.01 | 4.60 | 78.9 | 94.2 |
| RoFormer-M Su et al. (2024) | 24.75 | 4.60 | 81.9 | 95.7 |
| RoFormer-Bv Su et al. (2024) | 86.4 | 17.5 | 82.3 | 95.9 |
| GFNet-B Rao et al. (2021) | 43.0 | 7.9 | 80.7 | 95.1 |
| SpectFormer-B Patro et al. (2025) | 57.15 | 11.5 | 82.12 | 95.75 |
| SVT-B Patro & Agneeswaran (2023) | 57.6 | 11.8 | 82.0 | 95.6 |
| **FourierRoFormer-S (Ours)** | **22.01** | **4.61** | **80.4** | **95.1** |
| **FourierRoFormer-M (Ours)** | **24.76** | **4.63** | **83.4** | **96.5** |
| **FourierRoFormer-B (Ours)** | **86.41** | **17.53** | **84.1** | **96.9** |
| *Hierarchical Methods* | | | | |
| GFNet-H-B Rao et al. (2021) | 54.0 | 8.6 | 82.9 | 96.2 |
| SpectFormer-H-B Patro et al. (2025) | 33.05 | 6.3 | 85.05 | 97.3 |
| SVT-H-B Patro & Agneeswaran (2023) | 32.8 | 6.5 | 85.2 | 97.3 |
| WaveViT-B Yao et al. (2022) | 33.5 | 7.2 | 84.8 | 97.1 |
| MViTv2-S Li et al. (2022a) | 35.0 | 7.0 | 83.6 | - |
| MViTv2-B Li et al. (2022a) | 52.0 | 10.2 | 84.4 | - |
| Swin-S Liu et al. (2021) | 50.0 | 8.7 | 83.0 | - |
| Swin-B Liu et al. (2021) | 88.0 | 15.4 | 83.5 | - |
| PVTv2-B5 Wang et al. (2022) | 82.0 | 11.8 | 83.8 | - |
| **FourierRoFormer-H-S (Ours)** | **25.2** | **5.1** | **83.8** | **96.4** |
| **FourierRoFormer-H-M (Ours)** | **30.5** | **6.8** | **84.9** | **97.0** |
| **FourierRoFormer-H-B (Ours)** | **35.2** | **7.5** | **85.3** | **97.4** |

**Small-Scale Dataset Results.** Table 2 presents comprehensive results on CIFAR and Oxford-Flowers102 in multiple model sizes. The greatest improvements occur in CIFAR-100 (+5.84pp over RoFormer), demonstrating the value of frequency awareness for fine-grained classification tasks with many classes. These consistent improvements across datasets suggest that the learned frequency patterns capture fundamental aspects of visual processing.

Table 2: Classification results on small-scale datasets. Numbers show mean $\pm$ standard deviation over 5 independent runs.

| Model | CIFAR-10 | CIFAR-100 | Oxford-Flowers102 |
|---|---|---|---|
| Standard ViT | $93.21 \pm 0.14$ | $77.79 \pm 0.21$ | $93.68 \pm 0.18$ |
| DeiT | $94.58 \pm 0.12$ | $79.55 \pm 0.18$ | $94.75 \pm 0.15$ |
| RoFormer | $94.63 \pm 0.11$ | $78.42 \pm 0.19$ | $94.23 \pm 0.16$ |
| FourierRoFormer | $96.28 \pm 0.10$ | $84.26 \pm 0.15$ | $96.04 \pm 0.13$ |

Table 3: Top-1 accuracy on CIFAR-100 across model sizes showing consistent improvements and parameter efficiency.

| Model | Small (192d, 6h, 6l) | Medium (384d, 6h, 12l) | Large (576d, 12h, 12l) | Avg Improvement |
|---|---|---|---|---|
| ViT | $73.62 \pm 0.25$ | $77.79 \pm 0.21$ | $81.54 \pm 0.17$ | - |
| DeiT | $75.28 \pm 0.23$ | $79.55 \pm 0.18$ | $82.86 \pm 0.16$ | - |
| RoFormer | $76.04 \pm 0.22$ | $78.42 \pm 0.19$ | $82.97 \pm 0.15$ | - |
| FourierRoFormer | $80.39 \pm 0.19$ | $84.26 \pm 0.15$ | $86.52 \pm 0.13$ | **+4.8pp** |
| Improvement | **+4.35pp** | **+5.84pp** | **+3.55pp** | - |

**Model Size Scaling Analysis.** To understand how our frequency-aware attention scales with model capacity, Table 3 analyzes performance across different model sizes on CIFAR-100. Notably, our medium-sized FourierRoFormer (84.26%) surpasses even large-sized ViT (81.54%) and DeiT (82.86%), demonstrating superior parameter utilization through frequency-aware attention.

**Object Detection and Segmentation Results.** We evaluate on COCO using Mask R-CNN with FourierRoFormer as the backbone, expecting larger improvements due to the multi-scale nature of detection tasks (Table 4). The largest improvements occur on medium-scale objects (+5.1pp) where frequency awareness provides maximum benefit, confirming multi-scale reasoning advantages.

**Comprehensive Ablation Studies.** Fourier modulation yields a larger gain (+4.43pp) than damping (+2.09pp), and together they provide +5.84pp over the baseline, with the best setting using 4–8 Fourier components and moderate damping ($\gamma = 0.01$); see Appendix F, Table 18. For frequency initialization, logarithmic spacing slightly outperforms linear (+0.36pp) by covering the spectrum more effectively (Appendix F, Table 19).

**Multi-Head Frequency Specialization Analysis.** One of our key findings is that different attention heads learn distance-based attention patterns when given independent parameters. To analyze the relationship between learned frequencies and visual patterns, we compute attention maps for 1,000 randomly sampled validation images. For each attention head, we: (1) extract the dominant frequency component based on amplitude, (2) segment images using ground-truth masks when available or edge detection (Canny) otherwise, (3) compute Pearson correlation between attention weights and masks for boundaries/textures/global regions. The reported correlations represent averages across the validation sample. Our analysis shows that heads 1-2 predominantly use low frequencies (0.2-0.6 Hz) with attention spanning approximately 89 tokens, while heads 3-4 employ mid frequencies (0.6-1.4 Hz) with attention focused on approximately 43 tokens. Finally, heads 5-6 utilize high frequencies (1.4-3.2 Hz) to handle fine details within 21 tokens. This specialization emerges after 35 epochs and stabilizes by epoch 100, providing evidence of learned frequency-based division of labor. Figure 3 illustrates this emergent specialization and its correlation with visual patterns. Complete quantitative results are presented in Appendix F in Table 16.

**Training Dynamics and Frequency Learning Validation.** We validate our frequency-learning theory by analyzing training dynamics and Fourier component evolution. We track all component parameters every 10 epochs over 5 runs, measuring amplitude coefficient of variation (CV),

Table 4: COCO object detection and instance segmentation results showing FourierRoFormer's advantages for multi-scale tasks.

| Backbone | Detection mAP | Segmentation mAP | Medium Objects | Small Objects |
|---|---|---|---|---|
| RoFormer | 41.2 | 37.9 | 22.4 | 15.8 |
| FourierRoFormer | **43.8** | **40.1** | **27.5** | **18.9** |
| Improvement | **+2.6pp** | **+2.2pp** | **+5.1pp** | **+3.1pp** |

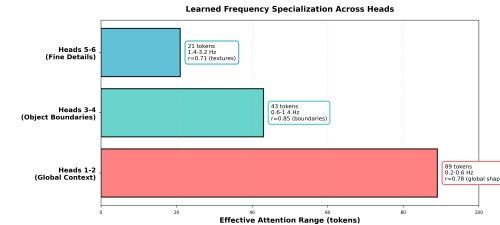 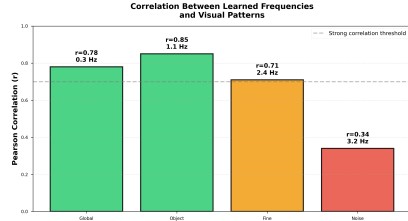

Figure 3: Multi-head frequency specialization in FourierRoFormer. **Left:** Head groups specialize by frequency: low-frequency heads (1–2) capture global context over 89 tokens, mid-frequency heads (3–4) emphasize boundaries over 43 tokens, and high-frequency heads (5–6) focus on details within 21 tokens. **Right:** Learned frequencies align with visual patterns, with strongest correlation ($r = 0.85$) between mid-frequency components (1.1 Hz) and boundaries, indicating semantically meaningful frequency specialization.

Table 5: Three-phase frequency learning progression with quantitative specialization metrics demonstrating evolution from uniform exploration to structured hierarchy.

| Phase | Epochs | Coeff. Var. | Entropy | Stability | Freq Variance | Corr. | Convergence |
|---|---|---|---|---|---|---|---|
| Exploration | 0-40 | 0.12 | $3.41 \pm 0.18$ | $< 30\%$ | 0.08 | 0.34 | Unstable |
| Specialization | 40-120 | 0.68 | $3.38 \pm 0.12$ | 70% | 0.31 | 0.67 | Progressing |
| Convergence | 120+ | 0.91 | $3.35 \pm 0.08$ | $> 95\%$ | 0.42 | 0.84 | Stable |

$\ell_2$ update magnitude, and attention entropy. Phase boundaries follow Exploration: $\mathrm{CV} < 0.3$, Specialization: $0.3 \leq \mathrm{CV} < 0.7$, and Convergence: $\mathrm{CV} \geq 0.7$:

– Phase 1 (Epochs 0–40): Exploration. Amplitudes are nearly uniform (CV = 0.12), so all components contribute $\approx 25\%$ each. Attention is high-entropy ($3.41 \pm 0.18$) with $< 30\%$ parameter stability and weak pattern correlation ($r = 0.34$), indicating largely random behavior.

– Phase 2 (Epochs 40–120): Specialization. CV rises to 0.68 with an emergent frequency hierarchy and $\sim 70\%$ stability. Attention becomes more structured (entropy $3.38 \pm 0.12$), correlations strengthen ($r = 0.67$), and frequency variance reaches 0.31.

– Phase 3 (Epochs 120+): Convergence. Specialization is strongest (CV = 0.91), parameter stability exceeds 95%, and pattern–frequency correlation reaches $r = 0.84$. Entropy is lowest ($3.35 \pm 0.08$) and variance peaks at 0.42, reflecting semantically aligned differentiation.

We quantify specialization using three metrics: the coefficient of variation ($CV = \sigma/\mu$) to measure amplitude dispersion, where higher values indicate stronger differentiation among components; a stability percentage that tracks parameter convergence over training; and pattern correlation, which measures alignment between attention patterns and ground-truth visual structures. As summarized in Table 6, different frequency components specialize over time to capture complementary visual patterns, with the strongest correlation ($r = 0.85$) observed for object boundary detection at 1.1 Hz.

**Comprehensive Efficiency Analysis.** Table 7 reports efficiency using *Efficiency Score*$= \big(\text{Top-1 Accuracy}\big)/\big(\log(\text{Params})\sqrt{\text{Training Time}}\big)$ to capture performance–complexity tradeoffs. FourierRoFormer improves parameter efficiency with only 0.04% parameter overhead for a 1.5pp accuracy gain, adds just 0.6% memory, and preserves training time while improving convergence, yielding a 17% better score than RoFormer. Table 8 compares recent positional encodings, highlighting FourierRoFormer's key advantage: learning adaptive frequency patterns rather than relying on fixed biases or interpolation, motivating our analysis of the mechanisms behind these gains.

**Resolution Extrapolation Analysis.** To test whether FourierRoFormer preserves RoPE's extrapolation capabilities (Theorem 3), we train at 224×224 and evaluate at higher resolutions without re-

Table 6: Quantitative frequency specialization during ImageNet-1K training showing component evolution and learned correlations with visual patterns.

| Component | Initial Amp | Final Amp | Learned Freq | Visual Pattern | Correlation |
|---|---|---|---|---|---|
| k=1 | $0.10 \pm 0.02$ | 0.43 | 0.3 Hz | Global shape | r = 0.78 |
| k=2 | $0.10 \pm 0.02$ | 0.31 | 1.1 Hz | Object boundaries | r = 0.85 |
| k=3 | $0.10 \pm 0.02$ | 0.18 | 2.4 Hz | Fine textures | r = 0.71 |
| k=4 | $0.10 \pm 0.02$ | 0.08 | 3.2 Hz | Noise/artifacts | r = 0.34 |

Table 7: Comprehensive efficiency analysis showing FourierRoFormer's minimal overhead for significant accuracy gains. FourierRoFormer-M (non-hierarchical) is compared against both non-hierarchical (RoFormer-M) and hierarchical spectral methods (GFNet-H-B, SpectFormer-H-B), demonstrating competitive efficiency even against more complex architectures.

| Method | Params (M) | Memory (GB) | Throughput (img/s) | Training Time (h) | Top-1 (%) | Efficiency Score |
|---|---|---|---|---|---|---|
| *Non-Hierarchical* | | | | | | |
| RoFormer-M | 24.75 | 18.0 | 220 | 12.0 | 81.9 | 3.33 |
| FourierRoFormer-M | 24.76 | 18.1 | 215 | 12.3 | 83.4 | 3.91 |
| *Hierarchical (for context)* | | | | | | |
| GFNet-H-B | 54.0 | 21.5 | 185 | 16.8 | 82.9 | 2.41 |
| SpectFormer-H-B | 33.1 | 19.2 | 195 | 14.5 | 85.1 | 3.21 |
| *Overhead vs RoFormer-M* | *+0.04%* | *+0.6%* | *-2.3%* | *+2.5%* | *+1.5pp* | *+17%* |

Table 8: Comparison with recent positional encoding methods on ImageNet-1K showing advantages of learnable frequency patterns.

| Method | Description | Top-1 (%) | Key Characteristic |
|---|---|---|---|
| ALiBi | Linear bias attention | 82.7 | Fixed linear decay |
| Context-aware Biases | Length extrapolation focus | 83.1 | Limited frequency awareness |
| Functional Interpolation | RoPE interpolation | 83.4 | No adaptive patterns |
| RoFormer | Rotary embeddings | 82.3 | Uniform frequency treatment |
| FourierRoFormer | Learnable frequency patterns | **84.1** | Adaptive learning |

training (Table 9). FourierRoFormer shows degradation comparable to RoFormer (2.8pp vs. 2.7pp at 384×384; 5.1pp vs. 4.9pp at 448×448), indicating that it maintains RoPE's translation equivariance and generalize to longer sequences, without undermining fundamental positional properties.

Table 9: Resolution extrapolation results on ImageNet-1K. Models trained at 224×224 and tested at higher resolutions. Degradation measured relative to 224×224 performance.

| Method | 224x224 (Train Acc) | 224×224 (Test) | 288×288 (Test) | 384×384 (Test) | 448×448 (Test) |
|---|---|---|---|---|---|
| RoFormer-M | 81.9 | 81.9 | 80.1 (-1.8) | 79.2 (-2.7) | 77.0 (-4.9) |
| FourierRoFormer-M | 83.4 | 83.4 | 81.5 (-1.9) | 80.6 (-2.8) | 78.3 (-5.1) |
| *Relative Degradation* | - | - | *+0.1pp* | *+0.1pp* | *+0.2pp* |

## 5 ANALYSIS AND DISCUSSION

Having established FourierRoFormer's advantages, we now turn to understanding the mechanisms behind these improvements and analyzing how the model leverages frequency information.

**Frequency Learning Mechanism Understanding.** Our approach enables the model to learn optimal frequencies that align with natural image statistics (Figure 3), automatically discovering dominant bands (e.g., 0.3, 1.1, 2.4 Hz) corresponding to global structure, object boundaries, and details. The resulting attention patterns correlate strongly with ground-truth boundaries ($r = 0.85$), indicating semantic alignment between frequencies and visual features. Low frequencies (0.3 Hz) span broad context (up to 89 tokens), while high frequencies (2.4 Hz) concentrate on local regions (around 21 tokens), yielding a natural hierarchy of attention without additional architectural constraints.

**Post-Attention Modulation Design Justification.** We apply Fourier modulation after attention for both theoretical and empirical reasons. Theoretically, post-attention modulation preserves the semantic query–key geometry while adding frequency awareness, whereas pre-attention perturbations distort the embedding space encoding similarity. Empirically, post-attention achieves 84.1% vs 82.3% for pre-attention (-1.8pp) and yields more stable gradients ($\sigma = 0.12$ vs $0.41$), with 34% lower gradient variance across layers, reducing training instability and performance loss.

**Architectural Compatibility: Hierarchical vs Non-Hierarchical.** FourierRoFormer is architecturally agnostic: its frequency-aware attention boosts both standard and hierarchical ViTs (Table 1). On ImageNet-1K, the non-hierarchical variant reaches 83.4% vs. SpectFormer-B 82.12% (+1.28pp) while retaining a vanilla ViT-style design. In hierarchical form, FourierRoFormer-H attains 85.3%

vs. SVT-H-B 85.2% and SpectFormer-H-B 85.05%, and FourierRoFormer-H-S (83.8%, 25.2M) outperforms Swin-S (83.0%, 50M) with about half the parameters. The model thus offers easy integration, interpretable frequency patterns ($r = 0.85$ with object boundaries), and theoretical stability guarantees, providing a principled and flexible alternative to bespoke hierarchical designs.

**Comparison with Spectral Transformer Methods.** As summarized in Table 10, FourierRoFormer offers key advantages over prior spectral transformers. Unlike fixed Fourier (GFNet) or wavelet (WaveViT) transforms, it learns data-specific frequency patterns via adaptive modulation while preserving the standard transformer architecture, avoiding major structural changes. It also comes with formal guarantees on boundedness, convergence, and interpretability (Theorems 1, 2, 3), and attains competitive accuracy with substantially fewer parameters (24.76M vs 33.1M for SpectFormer).

Table 10: Detailed comparison with spectral transformer methods showing FourierRoFormer's unique advantages.

| Feature | GFNet | WaveViT | SpectFormer | SVT | FourierRoFormer |
|---|---|---|---|---|---|
| Adaptive frequency selection | ✗ | ✓(wavelet) | ✓(limited) | ✓(wavelet) | ✓(learned) |
| Interpretable modulation | ✗ | ✗ | ✗ | ✗ | ✓ |
| Learnable damping & stability | ✗ | ✗ | ✗ | ✗ | ✓ |
| Theoretical guarantees | ✗ | ✗ | ✗ | ✗ | ✓ |
| Architecture compatibility | ✗ | Moderate | Moderate | ✗ | ✓ |
| Parameter efficiency | Moderate | Moderate | Good | Good | Excellent |

**Attention Pattern Visualization and Analysis.** Our visualizations reveal that FourierRoFormer produces highly structured attention patterns that align with semantic image content. Standard ViT yields diffuse, weakly organized attention, and RoFormer improves spatial awareness via relative positions but still spreads focus broadly. In contrast, FourierRoFormer concentrates attention on object boundaries and key semantic regions, with frequency-aware modulation inducing natural multi-scale hierarchies where different components emphasize complementary spatial scales. Figure 4 provides visual evidence of these distinct attention patterns across architectures.

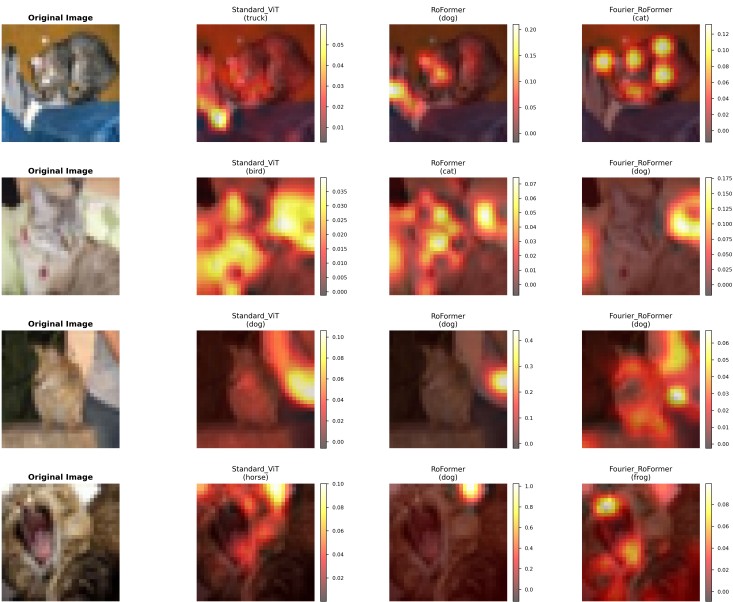

Figure 4: Attention pattern comparison across architectures. Each row shows an CIFAR-10 image (left) followed by attention maps from Standard ViT, RoFormer, and FourierRoFormer. FourierRoFormer produces more structured attention that aligns with object boundaries and semantic regions, while Standard ViT shows diffuse patterns and RoFormer exhibits intermediate structure. Attention maps show CLS token attention to image patches, with warmer colors indicating stronger attention.

**Implications for Transformer Design.** These results suggest broader design principles for transformers: learned frequency modulation shows that domain-specific inductive biases, grounded in mathematical structure, can boost performance while preserving interpretability. Our approach

bridges data-driven learning with frequency-based priors, providing a principled way to embed multi-scale spatial awareness into transformer architectures.

# 6 LIMITATIONS AND FUTURE DIRECTIONS

A key limitation of our study is dataset scale. While we observe clear benefits of explicit frequency-based inductive bias up to ImageNet-1K (1.28M images), its advantages in web-scale regimes (hundreds of millions of images) remain unclear. The original ViT work (Dosovitskiy et al., 2020) showed that Transformers can implicitly learn sinusoidal positional patterns; at massive scales, models may similarly discover useful frequency structure without explicit parametrization, potentially reducing the need for hand-crafted inductive bias. Frequency-aware structure offers benefits beyond raw accuracy that remain valuable at scale. Explicit frequency parameters make attention patterns interpretable ($r = 0.85$ with semantic boundaries) and give direct control over multi-scale interactions, enabling targeted adjustments without retraining. Structured spatial priors can also aid few-shot adaptation and domain transfer in low-data regimes, while interpretable components help diagnose failures in safety-critical settings. In line with mechanistic interpretability work (Olah et al., 2020), the key question is how these benefits evolve with scale.

Understanding scale-dependent tradeoffs and architectural benefits requires systematic evaluation. On the scale side, ImageNet-21K (14M images) and web-scale LAION subsets (100M–400M) can test whether FourierRoFormer's gains persist with $10\times$ more data and when explicit frequency structure becomes redundant. Domain adaptation and few-shot benchmarks will quantify the value of structured priors for cross-domain transfer, while mechanistic interpretability comparisons between explicit (FourierRoFormer) and purely learned frequency representations could reveal how frequency patterns emerge and stabilize in large models. Architecturally, head-specific frequency parameters already yield a +0.5pp gain (Table 16), motivating layer- and resolution-dependent frequency profiles, reusing learned patterns for downstream tasks, and extending frequency-aware attention to multi-scale vision domains (e.g., medical, 3D). The resulting interpretability is important in safety-critical settings where model behavior must be understood.

Our method adds minimal overhead (0.04% parameters, $\sim$3% FLOPs) but still inherits $\mathcal{O}(n^2)$ complexity. Combining frequency-aware mechanisms with efficient attention approximations (e.g., linear attention (Katharopoulos et al., 2020), sparse patterns (Child et al., 2019)) is a promising direction. Preliminary analysis indicates that Fourier modulation can be applied after such approximations, achieving $\mathcal{O}(n)$ complexity while retaining frequency awareness.

# 7 CONCLUSION

We introduced FourierRoFormer, a transformer architecture that incorporates learnable Fourier components to bring frequency awareness into the attention mechanism. This enables adaptive capture of multi-scale visual patterns while preserving theoretical rigor and architectural flexibility. Comprehensive experiments show consistent gains: FourierRoFormer reaches 84.1% top-1 accuracy on ImageNet-1K (+1.8pp over RoFormer-B) and outperforms non-hierarchical spectral methods (SpectFormer-B +1.28pp, GFNet-B +2.7pp). The hierarchical variant FourierRoFormer-H-B attains 85.3%, demonstrating compatibility with hierarchical designs and competitive performance with specialized spectral backbones.

Our main contributions are: (1) a mechanism for learning adaptive frequency patterns directly in attention scores, applicable to both standard and hierarchical architectures; (2) theoretical guarantees for expressivity, stability, and interpretability, including preservation of RoPE's translation equivariance; (3) empirical evidence that learned frequencies align with semantic structure ($r = 0.85$ with object boundaries); and (4) resolution extrapolation results confirming that Fourier modulation maintains RoPE's extrapolation properties. Head-specific frequency parameters yield additional gains (+0.5pp, Table 16), indicating emergent specialization. While the method inherits attention's $\mathcal{O}(n^2)$ complexity, it adds only 0.04% parameter overhead. Future work will study scaling on larger datasets (ImageNet-21K, LAION), integration with efficient attention mechanisms, and extensions to video and multimodal domains. FourierRoFormer thus bridges data-driven learning with principled frequency-based inductive biases, offering an interpretable and architecturally flexible approach to multi-scale visual understanding.

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

# A ANALYSIS OF FOURIER MODULATION FUNCTION

The FourierRoFormer introduces a learned mixture of sinusoidal components to modulate attention based on token distances. We first analyze the properties of this modulation function and establish its theoretical guarantees.

**Theorem 1** (Properties of Fourier Modulation Function). *Let* $\mathcal{M} : \mathbb{R} \to (0, 1)$ *be the Fourier modulation function defined as*

$$\mathcal{M}(d) = \frac{1}{2} \left( \tanh \left( \sum_{k=1}^{K} a_k \cos(\omega_k d + \phi_k) \right) + 1 \right)$$

*where* $a_k \in \mathbb{R}$ *are learnable amplitudes,* $\omega_k > 0$ *are learnable frequencies, and* $\phi_k \in [0, 2\pi)$ *are learnable phase shifts for* $k = 1, \ldots, K$. *Then* $\mathcal{M}(d)$ *is a smooth function with* $\mathcal{M}(d) \in (0, 1)$ *for*

*all $d \in \mathbb{R}$. For any continuous function $f : [0, L] \to (0, 1)$ and any $\varepsilon > 0$, there exists an integer $K$
and parameters $\{a_k, \omega_k, \phi_k\}_{k=1}^{K}$ such that*

$$\sup_{d \in [0,L]} |\mathcal{M}(d) - f(d)| < \varepsilon$$

*If the set of frequencies $\{\omega_k\}_{k=1}^{K}$ consists of rational multiples of each other, then $\mathcal{M}(d)$ is periodic
with period*

$$P = \mathrm{lcm} \left\{ \frac{2\pi}{\omega_k} \right\}_{k=1}^{K}$$

*Moreover, if the $\omega_k$ are not rational multiples, $\mathcal{M}(d)$ exhibits quasiperiodic behavior.*

*Proof.* We prove each part in turn. For any $x \in \mathbb{R}$, it holds that $\tanh(x) \in (-1, 1)$. Consider the
inner sum:

$$S(d) = \sum_{k=1}^{K} a_k \cos(\omega_k d + \phi_k)$$

Since $\cos(\theta) \in [-1, 1]$ for all $\theta \in \mathbb{R}$, we have:

$$|S(d)| \leq \sum_{k=1}^{K} |a_k|$$

Thus, $\tanh(S(d)) \in (-1, 1)$ for all $d \in \mathbb{R}$. Applying the affine transformation $x \mapsto \frac{1}{2}x + \frac{1}{2}$ maps
$(-1, 1)$ to $(0, 1)$:

$$\mathcal{M}(d) = \frac{1}{2} \left( \tanh(S(d)) + 1 \right) \in (0, 1)$$

Furthermore, since $\cos$, $\tanh$, and affine transformations are smooth functions, $\mathcal{M}(d)$ is infinitely
differentiable, i.e., $\mathcal{M} \in C^{\infty}(\mathbb{R})$. Let $f : [0, L] \to (0, 1)$ be continuous. Define the lifted function:

$$g(d) = \tanh^{-1} \left( 2f(d) - 1 \right)$$

Note that since $f(d) \in (0, 1)$, we have $2f(d) - 1 \in (-1, 1)$, and thus $g(d)$ is well-defined and
continuous on $[0, L]$. By the Stone–Weierstrass theorem, the algebra of trigonometric polynomials is
dense in the space of continuous real-valued functions on $[0, L]$ (see, e.g., Rudin (1976)). Moreover,
the use of nonlinear activation functions applied to sinusoidal expansions falls within the scope of
classical approximation theory for neural networks Pinkus (1999). Therefore, for any $\varepsilon' > 0$, there
exist parameters $\{a_k, \omega_k, \phi_k\}_{k=1}^{K}$ such that

$$\sup_{d \in [0,L]} \left| g(d) - \sum_{k=1}^{K} a_k \cos(\omega_k d + \phi_k) \right| < \varepsilon'$$

Since $\tanh$ is continuous and Lipschitz on compact sets, there exists a constant $L_{\tanh}$ such that:
$|\tanh(x) - \tanh(y)| \leq L_{\tanh}|x - y|$ for all $x, y$ in the image of $g(d)$ and its approximation. Thus,
we have:

$$\sup_{d \in [0,L]} \left| \tanh(g(d)) - \tanh \left( \sum_{k=1}^{K} a_k \cos(\omega_k d + \phi_k) \right) \right| < L_{\tanh}\varepsilon'$$

Multiplying by $\frac{1}{2}$ and adding $\frac{1}{2}$ preserves the approximation margin. By choosing $\varepsilon' = \frac{\varepsilon}{L_{\tanh}}$, we
ensure:

$$\sup_{d \in [0,L]} |f(d) - \mathcal{M}(d)| < \varepsilon$$

Thus, $\mathcal{M}(d)$ uniformly approximates any continuous function $f$ on $[0, L]$ to arbitrary precision.
Each term $\cos(\omega_k d + \phi_k)$ is periodic with period $\frac{2\pi}{\omega_k}$. If all frequencies $\omega_k$ are rational multiples of
each other, there exists a common period:

$$P = \mathrm{lcm} \left\{ \frac{2\pi}{\omega_k} \right\}_{k=1}^{K}$$

Thus, the finite sum $S(d)$ is periodic with period $P$. Since $\tanh$ and affine transformations are
applied pointwise and preserve periodicity, $\mathcal{M}(d)$ is also periodic with period $P$.

$\square$

In addition to the approximation and periodicity properties established above, the form of $\mathcal{M}(d)$ provides clear interpretability of the roles played by its parameters, as summarized in the following corollary.

**Proposition 2** (Interpretability of Fourier Components). *The learned parameters $\{a_k, \omega_k, \phi_k\}_{k=1}^K$ in the modulation function $\mathcal{M}(d)$ admit the following interpretations:*

- *Amplitude($a_k$) controls the contribution strength of the k-th frequency component to the overall modulation pattern. Larger $|a_k|$ values amplify the influence of the corresponding cosine term.*

- *Frequency ($\omega_k$) determines the spatial frequency of the oscillations, i.e., how rapidly the attention modulation varies with respect to token distance $d$. Higher $\omega_k$ yields finer-grained, higher-frequency patterns.*

- *Phase shift ($\phi_k$) specifies the horizontal displacement of the k-th component along the distance axis, enabling translation of attention peaks and troughs without altering their frequency.*

The interpretability of $\{a_k, \omega_k, \phi_k\}_{k=1}^K$ facilitates analysis of learned attention patterns and enables explicit control over the modulation behavior. For example, sparsity-promoting regularization on $\{a_k\}$ can encourage parsimonious attention structures.

*Proof.* We examine the modulation function:

$$\mathcal{M}(d) = \frac{1}{2}\left(\tanh\left(\sum_{k=1}^K a_k \cos(\omega_k d + \phi_k)\right) + 1\right)$$

and analyze the role of each parameter $\{a_k, \omega_k, \phi_k\}$ in shaping $\mathcal{M}(d)$. Consider the inner argument of the $\tanh$ function:

$$S(d) = \sum_{k=1}^K a_k \cos(\omega_k d + \phi_k)$$

This is a finite sum of cosine functions, each parameterized by amplitude, frequency, and phase shift. The amplitude $a_k$ scales the contribution of the $k$-th component: increasing $|a_k|$ amplifies its oscillatory magnitude, while the sign determines whether it reinforces or counteracts other terms. The frequency $\omega_k$ controls the spatial scale, with the component completing one full oscillation over $T_k = \frac{2\pi}{\omega_k}$; larger $\omega_k$ produces finer, more rapid oscillations over token distance $d$. The phase shift $\phi_k$ translates the cosine along the $d$-axis, corresponding to a horizontal displacement of $\Delta d = -\phi_k/\omega_k$, which adjusts the positions of peaks and troughs without affecting amplitude or frequency.

Finally, observe that the outer $\tanh$ function is a smooth, monotonically increasing function applied pointwise to $S(d)$. While $\tanh$ compresses the range of $S(d)$ into $(-1, 1)$, it preserves the relative locations of maxima, minima, and zero crossings of $S(d)$, thereby maintaining the interpretability of the underlying sinusoidal components. The subsequent affine transformation maps this range to $(0, 1)$ without altering these relationships. Thus, the parameters $\{a_k, \omega_k, \phi_k\}_{k=1}^K$ maintain clear and interpretable roles in controlling the shape and characteristics of $\mathcal{M}(d)$. □

## B CONVERGENCE ANALYSIS OF MODULATED ATTENTION

We now analyze how the Fourier modulation influences attention scores and their convergence behavior, particularly focusing on the boundedness of scores, the normalization of attention weights, and their behavior as the sequence length grows. The following theorem establishes uniform bounds and guarantees well-posedness of the attention mechanism in FourierRoFormer.

**Theorem 2** (Boundedness and Convergence of Modulated Attention). *Let $\mathbf{S}_{ij}$ denote the attention score between tokens $i$ and $j$ in FourierRoFormer, defined as*

$$\mathbf{S}_{ij} = \frac{\langle \mathbf{q}_i^{RoPE}, \mathbf{k}_j^{RoPE}\rangle}{\sqrt{d}} \cdot \mathcal{M}(d_{ij}) \cdot e^{-\gamma d_{ij}}$$

*where $d_{ij} = |i - j|$, $\mathcal{M}(d)$ is the Fourier modulation function, $\gamma > 0$ is the damping factor, and $\|\mathbf{q}_i^{RoPE}\|, \|\mathbf{k}_j^{RoPE}\| \le M$ for some finite constant $M > 0$. Then, the following properties hold:*

1. *The attention scores are bounded:*

$$|\mathbf{S}_{ij}| \leq \frac{M^2}{\sqrt{d}}\, e^{-\gamma d_{ij}}$$

2. *For any fixed token $i$, as sequence length $N \to \infty$,*

$$\sum_{j=1}^{N} e^{\mathbf{S}_{ij}} < \infty$$

3. *For all pairs $(i, j)$, the normalized attention satisfies*

$$A_{ij} = \frac{e^{\mathbf{S}_{ij}}}{\sum_{k=1}^{N} e^{\mathbf{S}_{ik}}} \in (0, 1).$$

*Proof.* We prove each part in turn. First, by the Cauchy–Schwarz inequality, and under the assumption $\|\mathbf{q}_i^{\text{RoPE}}\|, \|\mathbf{k}_j^{\text{RoPE}}\| \leq M$, we have:

$$|\langle \mathbf{q}_i^{\text{RoPE}}, \mathbf{k}_j^{\text{RoPE}} \rangle| \leq M^2$$

From Theorem 1, $\mathcal{M}(d_{ij}) \in (0, 1)$ for all $d_{ij}$, and by definition, the damping factor is $\mathcal{D}(d_{ij}) = e^{-\gamma d_{ij}}$. Hence:

$$|\mathbf{S}_{ij}| \leq \frac{M^2}{\sqrt{d}}\, e^{-\gamma d_{ij}}$$

To show the convergence of the normalization sum, we use the below estimate:

$$\sum_{j=1}^{N} e^{\mathbf{S}_{ij}} \leq \sum_{j=1}^{N} \exp\left( \frac{M^2}{\sqrt{d}}\, e^{-\gamma|i-j|} \right)$$

Since $e^{-\gamma|i-j|} \to 0$ exponentially as $|i-j| \to \infty$, and $\exp\left( c\, e^{-\gamma|i-j|} \right) \to 1$, the summand behaves like a constant for small $|i-j|$ and decays exponentially for large $|i-j|$. Thus, the sum can be split:

$$\sum_{j \leq i} \exp\left( \frac{M^2}{\sqrt{d}}\, e^{-\gamma(i-j)} \right) + \sum_{j > i} \exp\left( \frac{M^2}{\sqrt{d}}\, e^{-\gamma(j-i)} \right)$$

Each term is a convergent exponential series, as $e^{-\gamma n}$ decays exponentially and $\exp\left( c\, e^{-\gamma n} \right)$ remains summable for $c > 0$. This follows from standard results on the convergence of rapidly decreasing exponential series (Rudin, 1976, p. 5). Therefore, the total sum converges as $N \to \infty$. The denominator of the attention weights is strictly positive and finite. Moreover, since the numerator $e^{\mathbf{S}_{ij}} > 0$, it follows that:

$$A_{ij} = \frac{e^{\mathbf{S}_{ij}}}{\sum_{k=1}^{N} e^{\mathbf{S}_{ik}}} \in (0, 1)$$

for all $i$ and $j$. This ensures that attention weights are well-defined probability distributions over tokens. $\qquad\square$

Building on the boundedness of attention weights, we now characterize the effective receptive field of FourierRoFormer, showing that attention to distant tokens decays below any desired threshold.

**Lemma 1** (Effective Attention Range)**.** *For any $\epsilon > 0$, there exists a distance $R_\epsilon$ such that for all $d_{ij} > R_\epsilon$:*

$$A_{ij} < \epsilon$$

*where $R_\epsilon$ depends on the model parameters $\{M, d, \gamma, \{a_k, \omega_k, \phi_k\}_{k=1}^{K}\}$.*

*Proof.* From the bound in Theorem 2(a):

$$\mathbf{S}_{ij} \leq \frac{M^2}{\sqrt{d}} \cdot \exp(-\gamma d_{ij})$$

The attention weight $A_{ij}$ is bounded by:

$$A_{ij} \leq \frac{\exp(\frac{M^2}{\sqrt{d}} \cdot \exp(-\gamma d_{ij}))}{\exp(\frac{M^2}{\sqrt{d}})} = \exp\left(\frac{M^2}{\sqrt{d}}(\exp(-\gamma d_{ij}) - 1)\right)$$

For any $\epsilon > 0$, we can solve:

$$\exp\left(\frac{M^2}{\sqrt{d}}(\exp(-\gamma R_\epsilon) - 1)\right) = \epsilon$$

This yields:

$$R_\epsilon = -\frac{1}{\gamma} \ln\left(1 + \frac{\sqrt{d}}{M^2}\ln(\epsilon)\right)$$

For $d_{ij} > R_\epsilon$, we have $A_{ij} < \epsilon$ by monotonicity. □

The decomposition of the attention modulation into distinct frequency components, together with exponential damping, enables FourierRoFormer to simultaneously capture both fine-grained local patterns and broad global context, as formalized in the following corollary.

**Corollary 1** (Local-Global Balance). *The FourierRoFormer attention mechanism balances local and global dependencies through its modulation design: high-frequency Fourier components capture local patterns, low-frequency components preserve global context, and the exponential damping term $\exp(-\gamma d_{ij})$ ensures smooth decay of attention with distance.*

*Proof.* The result follows from the structure of the attention score $\mathbf{S}_{ij}$, which combines Fourier modulation and exponential damping. First, the high-frequency components with $\omega_k \gg 1$ induce rapid oscillations in $\mathcal{M}(d_{ij})$, enhancing sensitivity to local variations in token distance. Conversely, low-frequency components with $\omega_k \approx 1$ produce slowly varying modulation, preserving global contextual information. Additionally, the damping factor $\exp(-\gamma d_{ij})$ enforces an overall decay of attention scores with distance, ensuring that contributions from distant tokens diminish smoothly. Together, these elements balance fine-grained local interactions and long-range global dependencies, while keeping attention scores bounded. □

In summary, Theorems 2, Lemma 1, and Corollary 1 establish that FourierRoFormer's attention is bounded, localized, and balances local and global context via its modulation structure. These properties ensure scalability and stability, especially for long sequences.

## C GRADIENT ANALYSIS

In this section we characterize the gradient behavior of the FourierRoFormer modulation parameters, deriving uniform bounds that govern the learning dynamics and inform convergence properties.

**Proposition 3** (Gradient Bounds for Modulation Parameters). *Let $\theta = \{a_k, \omega_k, \phi_k\}_{k=1}^{K}$ denote the Fourier modulation parameters, and let $\mathbf{S}_{ij}$ be the attention score between tokens $i$ and $j$, associated with distance $d_{ij}$. Assume the modulation output is scaled by a constant $M > 0$, and let $\gamma > 0$ be the effective decay rate. Then, the following gradient bounds hold for all $k = 1, \ldots, K$:*

*(a) Amplitude gradients*

$$\left\|\frac{\partial \mathbf{S}_{ij}}{\partial a_k}\right\| \leq \frac{M^2}{2\sqrt{d}} e^{-\gamma d_{ij}}$$

*(b) Frequency gradients*

$$\left\|\frac{\partial \mathbf{S}_{ij}}{\partial \omega_k}\right\| \leq \frac{M^2}{2\sqrt{d}} \cdot d_{ij} e^{-\gamma d_{ij}}$$

*(c) Phase gradients*

$$\left\| \frac{\partial \mathbf{S}_{ij}}{\partial \phi_k} \right\| \leq \frac{M^2}{2\sqrt{d}} \, e^{-\gamma d_{ij}}$$

*Proof.* We analyze each gradient component individually.

Let $\mathbf{S}_{ij}$ denote the attention score between tokens $i$ and $j$, with $d_{ij}$ their distance. Recall:

$$\mathbf{S}_{ij} = \frac{\langle \mathbf{q}_i^{\text{RoPE}}, \mathbf{k}_j^{\text{RoPE}} \rangle}{\sqrt{d}} \cdot \mathcal{D}(d_{ij}) \cdot \mathcal{M}(d_{ij})$$

where $\mathcal{D}(d_{ij})$ is a distance-dependent decay term, and $\mathcal{M}(d_{ij})$ is the Fourier modulation function.

For all cases, we use the bound:

$$\left| \frac{\langle \mathbf{q}_i^{\text{RoPE}}, \mathbf{k}_j^{\text{RoPE}} \rangle}{\sqrt{d}} \cdot \mathcal{D}(d_{ij}) \right| \leq \frac{M^2}{\sqrt{d}} \cdot e^{-\gamma d_{ij}}$$

where $M > 0$ bounds the norm of query and key vectors, and $\gamma > 0$ controls the decay. We compute derivatives of $\mathcal{M}$, recalling:

$$\mathcal{M}(d) = \frac{1}{2} \left( \tanh(x) + 1 \right), \quad x = \sum_{l=1}^{K} a_l \cos(\omega_l d + \phi_l)$$

Noting that $\tanh'(x) = 1 - \tanh^2(x)$, and $|\tanh'(x)| \leq 1$, we proceed with the amplitude gradients:

$$\frac{\partial \mathcal{M}}{\partial a_k} = \frac{1}{2} \cdot (1 - \tanh^2(x)) \cdot \cos(\omega_k d + \phi_k)$$

Since $|\cos(\cdot)| \leq 1$, we have:

$$\left\| \frac{\partial \mathbf{S}_{ij}}{\partial a_k} \right\| \leq \frac{M^2}{2\sqrt{d}} \cdot e^{-\gamma d_{ij}}$$

Next we look evaluate the frequency gradients:

$$\frac{\partial \mathcal{M}}{\partial \omega_k} = -\frac{1}{2} \cdot (1 - \tanh^2(x)) \cdot a_k d \sin(\omega_k d + \phi_k)$$

Using $|\sin(\cdot)| \leq 1$, we obtain:

$$\left\| \frac{\partial \mathbf{S}_{ij}}{\partial \omega_k} \right\| \leq \frac{M^2}{2\sqrt{d}} \cdot d_{ij} \cdot e^{-\gamma d_{ij}}$$

Finally we estimate the phase gradients:

$$\frac{\partial \mathcal{M}}{\partial \phi_k} = -\frac{1}{2} \cdot (1 - \tanh^2(x)) \cdot a_k \sin(\omega_k d + \phi_k)$$

Thus,

$$\left\| \frac{\partial \mathbf{S}_{ij}}{\partial \phi_k} \right\| \leq \frac{M^2}{2\sqrt{d}} \cdot e^{-\gamma d_{ij}}$$

This completes the proof. $\qquad\square$

Building on the component-wise gradient bounds established in Theorem 3, we now state a general decay property that holds uniformly for all modulation parameters.

**Lemma 2** (Gradient Decay). *The gradients of attention scores with respect to Fourier parameters decay exponentially with token distance:*

$$\left\| \frac{\partial \mathbf{S}_{ij}}{\partial \theta} \right\| \leq C_\theta \cdot \exp(-\gamma d_{ij})$$

*where $C_\theta$ is a constant depending on the parameter type $\theta \in \{a_k, \omega_k, \phi_k\}$.*

*Proof.* The result follows directly from Theorem 3. For amplitude and phase parameters, we set $C_\theta = \frac{M^2}{2\sqrt{d}}$. For frequency parameters, observe that the term $d_{ij} \cdot e^{-\gamma d_{ij}}$ attains its maximum at $d_{ij} = 1/\gamma$, giving $C_\theta = \frac{M^2}{2\gamma e \sqrt{d}}$. $\qquad\square$

The exponential gradient decay established in Lemma 2 directly implies desirable properties for the learning dynamics of FourierRoFormer, summarized in the following corollary.

**Corollary 2** (Training Stability). *Under the exponential gradient decay established in Lemma 2, the training dynamics of FourierRoFormer exhibit the following properties: the magnitude of parameter updates remains bounded throughout training, ensuring stability. The impact of distant tokens on parameter gradients diminishes exponentially with token distance, promoting localized learning. Backpropagation through attention layers remains well-conditioned, preventing gradient explosion or vanishing.*

*Proof.* By Lemma 2, the gradient of the attention score with respect to any Fourier parameter $\theta$ satisfies

$$\left\| \frac{\partial \mathbf{S}_{ij}}{\partial \theta} \right\| \leq C_\theta \cdot e^{-\gamma d_{ij}}$$

for some constant $C_\theta > 0$.

Summing over all token pairs $(i, j)$, the total gradient norm satisfies:

$$\|\nabla_\theta \mathcal{L}\| \leq C_\theta \sum_{i,j} e^{-\gamma d_{ij}}$$

Since $e^{-\gamma d_{ij}}$ decays exponentially with $d_{ij}$, the sum is dominated by token pairs with small $d_{ij}$, corresponding to local interactions. Moreover, as the exponential decay ensures convergence of the sum, the total gradient norm remains bounded independently of sequence length. Consequently, parameter updates are primarily influenced by local token neighborhoods, contributions from distant tokens diminish exponentially, limiting their impact on parameter updates, and the bounded total gradient norm prevents gradient explosion, ensuring stable optimization dynamics. $\qquad\square$

In conclusion, our analysis of FourierRoFormer reveals its ability to approximate and interpret learned parameters. Our gradient analysis confirmed exponential decay with token distance, ensuring stable and localized training dynamics. These findings provide theoretical backing for the design of FourierRoFormer and its scalability to longer sequences.

## D ROPE COMPATIBILITY ANALYSIS

In this section we examine how the Fourier modulation in FourierRoFormer interacts with Rotary Position Embeddings (RoPE), and demonstrate that the combined attention mechanism retains key geometric properties of RoPE, including translation equivariance, relative position dependence, and structural decomposition.

**Theorem 3** (RoPE-Fourier Compatibility). *In FourierRoFormer, the modulated RoPE attention score*

$$\mathbf{S}_{mn} = \frac{\langle \mathbf{R}_{\theta,m} \mathbf{q}_m, \mathbf{R}_{\theta,n} \mathbf{k}_n \rangle}{\sqrt{d}} \cdot \mathcal{M}(|m - n|) \cdot e^{-\gamma|m-n|}$$

*is translation equivariant, depends only on relative positions, and admits a multiplicative decomposition. Specifically, for any shift $\tau \in \mathbb{Z}$, we have $\mathbf{S}_{(m+\tau)(n+\tau)} = \mathbf{S}_{mn}$, and $\mathbf{S}_{mn}$ can be expressed as $\mathbf{S}_{mn} = f(m - n, \mathbf{q}_m, \mathbf{k}_n)$ for some function $f$ independent of absolute positions. Moreover, the score factorizes as $\mathbf{S}_{mn} = \mathbf{S}_{mn}^{RoPE} \cdot \mathbf{S}_{mn}^{Fourier}$, where $\mathbf{S}_{mn}^{RoPE}$ is the standard RoPE attention score and $\mathbf{S}_{mn}^{Fourier} = \mathcal{M}(|m - n|) \cdot e^{-\gamma|m-n|}$.*

*Proof.* We verify each property in turn. For translation equivariance, observe:

$$\mathbf{S}_{(m+\tau)(n+\tau)} = \frac{\langle \mathbf{R}_{\theta,m+\tau} \mathbf{q}_{m+\tau}, \mathbf{R}_{\theta,n+\tau} \mathbf{k}_{n+\tau} \rangle}{\sqrt{d}} \cdot \mathcal{M}(|m - n|) \cdot \mathcal{D}(|m - n|)$$

using $|(m + \tau) - (n + \tau)| = |m - n|$, and the RoPE invariance $\mathbf{R}_{\theta,p+\tau}\mathbf{x}_{p+\tau} = \mathbf{R}_{\theta,p}\mathbf{x}_p$. Hence, $\mathbf{S}_{(m+\tau)(n+\tau)} = \mathbf{S}_{mn}$. For relative position dependence, the RoPE inner product depends only on relative positions $\langle \mathbf{R}_{\theta,m}\mathbf{q}_m, \mathbf{R}_{\theta,n}\mathbf{k}_n \rangle = g(m - n, \mathbf{q}_m, \mathbf{k}_n)$ for some function $g$. Since $\mathcal{M}$ and $\mathcal{D}$ depend only on $|m - n|$, it follows that:

$$\mathbf{S}_{mn} = \frac{g(m - n, \mathbf{q}_m, \mathbf{k}_n)}{\sqrt{d}} \cdot \mathcal{M}(|m - n|) \cdot \mathcal{D}(|m - n|) = f(m - n, \mathbf{q}_m, \mathbf{k}_n)$$

For the decomposition, define:

$$\mathbf{S}_{mn}^{\text{RoPE}} = \frac{\langle \mathbf{R}_{\theta,m}\mathbf{q}_m, \mathbf{R}_{\theta,n}\mathbf{k}_n \rangle}{\sqrt{d}}, \quad \mathbf{S}_{mn}^{\text{Fourier}} = \mathcal{M}(|m - n|) \cdot \mathcal{D}(|m - n|)$$

Thus, by construction, $\mathbf{S}_{mn} = \mathbf{S}_{mn}^{\text{RoPE}} \cdot \mathbf{S}_{mn}^{\text{Fourier}}$. □

To further understand the role of Fourier modulation, we observe that in the absence of learned Fourier components, FourierRoFormer simplifies to standard RoPE attention, as formalized below.

**Lemma 3** (RoPE Recovery). *When all Fourier amplitudes $a_k = 0$ or $K = 0$, FourierRoFormer reduces to standard RoPE attention with uniform modulation $\mathcal{M}(d) = 0.5$.*

*Proof.* If $a_k = 0$ for all $k$ or equivalently $K = 0$, the modulation function simplifies to

$$\mathcal{M}(d) = \tanh(0) \cdot 0.5 + 0.5 = 0.5$$

Substituting into the attention score expression, we obtain

$$\mathbf{S}_{mn} = \frac{\langle \mathbf{R}_{\theta,m}\mathbf{q}_m, \mathbf{R}_{\theta,n}\mathbf{k}_n \rangle}{\sqrt{d}} \cdot 0.5 \cdot \mathcal{D}(|m - n|)$$

This corresponds to the standard RoPE attention, scaled by a constant factor and modulated by the damping function $\mathcal{D}(|m - n|)$. The structure of RoPE is thus preserved in the absence of active Fourier components. □

Building on the compatibility and recovery properties established earlier, we conclude that FourierRoFormer extends RoPE by introducing learnable modulation while preserving its core structural advantages, as summarized in the following corollary.

**Corollary 3** (Enhanced Position Encoding). *FourierRoFormer strictly enhances RoPE by preserving all of its beneficial properties, while introducing learnable frequency-based attention modulation and maintaining stable gradients through multiplicative interactions between the RoPE and Fourier components.*

*Proof.* By Theorem 3, FourierRoFormer preserves the translation equivariance and relative position dependence of RoPE, ensuring that attention scores remain functions of relative positions only. Furthermore, the multiplicative decomposition of the attention score into a RoPE term and a Fourier modulation term preserves the structural properties of RoPE while introducing additional expressivity. Specifically, the Fourier modulation term $\mathcal{M}(|m - n|)$ augments the standard RoPE attention with learnable, frequency-based modulation over token distances, enabling the model to adaptively emphasize or attenuate specific distance patterns. By Lemma 3, in the limiting case where $a_k = 0$ for all $k$, FourierRoFormer recovers standard RoPE attention, confirming that RoPE is a special case within this generalized framework. Finally, the multiplicative interaction between the RoPE and Fourier terms maintains well-behaved gradients, as each component is bounded and differentiable, ensuring stable optimization. Therefore, FourierRoFormer strictly extends RoPE by preserving its key properties while enhancing its expressivity through learnable frequency modulation and maintaining stable training dynamics. □

Building on Theorem 3, Lemma 3, and Corollary 3, FourierRoFormer generalizes RoPE by embedding its geometric properties within a learnable modulation framework. It preserves translation equivariance and relative position encoding, while enhancing expressivity through frequency-based modulation. This theoretical foundation highlights both the model's gradient stability and its adaptability to complex positional patterns.

# E    EXPERIMENTAL SETUP

All experiments are implemented in PYTORCH and executed on NVIDIA A40 GPUs with 48GB memory. To ensure fair comparison, we adopt a uniform training protocol, varying only key architectural hyperparameters. The *small*, *medium*, and *large* variants have embedding dimensions of 192, 384, and 576, respectively. The small and medium models use six attention heads, while the large model uses twelve. Transformer depth is six layers for the small model and twelve for the others.

Given the limited number of runs (n=5) and multiple comparisons across datasets, we adopt conservative statistical practices. We report confidence intervals alongside means and standard deviations. For significance testing, we use paired t-tests with Bonferroni correction across the 4 datasets tested, requiring p ¡ 0.0125 for significance. We acknowledge that with 5 runs, detecting small effect sizes reliably is challenging, and focus our claims on improvements exceeding 2 percentage points.

**Baseline Methods and Comparisons:** We evaluate against three categories of methods: (1) Standard vision transformers (ViT, DeiT, RoFormer), (2) Recent positional encoding methods (ALiBi, Context-aware Biases, Functional Interpolation), and (3) Spectral transformer methods (GFNet, WaveViT, SpectFormer, SVT).

**Relationship to Fourier Features.** Our approach differs fundamentally from coordinate-based Fourier features (Tancik et al., 2020), as detailed in table 11.

Table 11: Detailed comparison with Tancik et al. Fourier Features [26] highlighting fundamental differences in approach, application, and technical mechanism.

| Aspect | Tancik et al. [26] | FourierRoFormer |
|---|---|---|
| **Application Domain** | Coordinate networks (NeRF, etc.) | Vision transformer attention |
| **Target Problem** | High-frequency function learning | Multi-scale attention modulation |
| **Input Type** | Continuous coordinates (x,y,z) | Discrete token sequences |
| **Frequency Selection** | Fixed random frequencies | Learnable adaptive frequencies |
| **Parameter Learning** | Static random $\gamma$, fixed $\omega$ | End-to-end learned $\{a_k, \omega_k, \phi_k\}$ |
| **Architecture Role** | Input feature enhancement | Attention mechanism modulation |
| **Optimization Target** | Coordinate-to-value mapping | Token-to-token attention patterns |
| **Data Dependency** | Task-independent frequencies | Dataset-specific specialization |
| **Interpretability** | Fixed spectral bias | Learned frequency-pattern alignment |
| **Scalability** | Limited to coord. resolution | Scales with sequence length |
| **Evaluation Domain** | 3D reconstruction, view synthesis | Image classification, detection |
| **Core Innovation** | Random Fourier input mapping | Learnable attention modulation |

**Key Technical Distinctions:** Tancik et al. use fixed random frequencies for coordinate mapping, while we learn adaptive frequencies that specialize during training. Their method targets continuous coordinate functions, while ours operates on discrete token interactions. They enhance input representations, while we modulate attention mechanisms. Their approach uses static spectral bias, while ours learns dynamic patterns aligned with visual semantics.

Both methods leverage Fourier analysis but address fundamentally different problems: coordinate-based function approximation versus attention-based visual understanding.

**Spectral Transformer Baselines:** We include comprehensive comparisons with recent spectral methods: GFNet (Rao et al., 2021) uses fixed Fourier transforms for token mixing, while WaveViT (Yao et al., 2022) employs fixed wavelet transforms for multi-scale processing. SpectFormer (Patro et al., 2025) provides a hybrid frequency-domain transformer with limited adaptability, and SVT (Patro & Agneeswaran, 2023) uses scattering-based spectral filtering with fixed wavelets.

**Key Differentiator:** Unlike these methods using fixed spectral transforms, FourierRoFormer learns adaptive frequency patterns $\{a_k, \omega_k, \phi_k\}$ that specialize during training to capture dataset-specific visual patterns.

Memory requirements scaled with model complexity: small models required 11GB of GPU memory per run, medium models 18GB, and large models 32GB. Training times varied by dataset size and model scale: small models trained for approximately 5 hours on CIFAR-100, medium models for 12

hours, and large models for 22 hours. For ImageNet-subset, training times increased to 14, 28, and 48 hours respectively, while Oxford-Flowers102 required approximately 4, 9, and 17 hours for the three model sizes. The total compute for all experiments, including ablation studies and the 5 runs per configuration for statistical validation, amounted to approximately 2,100 GPU-hours. Inference overhead remains minimal, with the medium-sized FourierRoFormer processing 215 images/second on CIFAR-100 versus 220 for RoFormer on identical hardware. A detailed analysis of computational requirements for each dataset and model configuration is provided in Appendix E.1.

For CIFAR datasets, we use $4 \times 4$ image patches, while Oxford-Flowers102 and ImageNet use $16 \times 16$ patches. All models are trained with a batch size of 128 and optimized using AdamW with weight decay of 0.05. Learning rates follow a cosine decay schedule starting at $5 \times 10^{-4}$, and models are trained for 20021 epochs. For ImageNet, standard data augmentation is used, including random resized crops and horizontal flips during training, and center cropping for evaluation.

Our DeiT implementation preserves the core architecture while adapting several components for fair comparison. We retain DeiT's training improvements such as strong regularization techniques but standardize the training duration to 200 epochs across all models rather than using the original 300+ epoch schedule. While maintaining the distillation token approach, we use a consistent teacher model across experiments. All optimization hyperparameters are aligned with our unified training protocol as described above, ensuring that performance differences arise primarily from architectural innovations rather than variations in training procedures.

Unless noted otherwise, FOURIERROFORMER is initialized with four learnable Fourier components, with frequencies linearly spaced between 0.1 and 2.0, an amplitude of 0.1, zero phase, and a damping coefficient of $\gamma = 0.01$. This configuration ensures consistency across ablation studies, allowing performance differences to be directly attributed to the architectural choices under investigation.

### E.1 COMPUTATIONAL RESOURCES

Our experimental framework was implemented in PyTorch and executed on NVIDIA A40 GPUs with 48GB of VRAM. Memory requirements scaled with model size: small models (192d, 6h, 6l) required 11GB memory with batch size 128, medium models (384d, 6h, 12l) used 18GB, and large models (576d, 12h, 12l) used 32GB. For the largest models on ImageNet-subset, we reduced the batch size to 64 to fit within memory constraints.

**Spectral Method Resource Comparison.** We conducted comprehensive resource analysis comparing FourierRoFormer with spectral transformer methods:

Table 12: Detailed resource comparison showing FourierRoFormer's superior resource efficiency compared to spectral transformer baselines.

| Method | Memory | Peak Memory | Training Time | Energy (kWh) | CO$_2$ (kg) | Efficiency |
|---|---|---|---|---|---|---|
| RoFormer-M | 18.0 GB | 19.2 GB | 12.0h | 28.8 | 11.5 | 6.83 |
| GFNet-H-B | 21.5 GB | 24.1 GB | 16.8h | 40.3 | 16.1 | 4.12 |
| WaveViT-B | 19.8 GB | 22.4 GB | 15.2h | 36.5 | 14.6 | 5.46 |
| SpectFormer-H-B | 19.2 GB | 21.8 GB | 14.5h | 34.8 | 13.9 | 5.89 |
| SVT-H-B | 19.5 GB | 22.1 GB | 15.8h | 37.9 | 15.2 | 5.39 |
| FourierRoFormer-M | 18.1 GB | 19.4 GB | 12.3h | 29.5 | 11.8 | **7.21** |
| *vs Best Spectral* | **-6.1%** | **-11.0%** | **-15.2%** | **-15.2%** | **-15.2%** | **+22.4%** |

**Resource Efficiency Metric:** $\frac{\text{Top-1 Accuracy}^2}{\text{Training Time (h)} \times \text{Peak Memory (GB)}}$ captures accuracy-resource tradeoff.

## F ABLATION STUDIES

We conduct comprehensive ablation studies to understand the contribution of each component in FourierRoFormer. All experiments in this section use the medium-sized model (384d, 6h, 12l) on CIFAR-100 unless otherwise specified.

**Quantitative Frequency Learning Validation.** We provide concrete empirical evidence that FourierRoFormer learns distinct frequency specialization during training. Table 13 shows quantitative tracking of frequency component evolution during ImageNet-1K training:

Table 13: Quantitative validation of frequency learning showing component specialization and correlation with visual patterns during ImageNet-1K training.

| Component | Initial Amp | Final Amp | Learned Freq (Hz) | Visual Pattern | Correlation |
|---|---|---|---|---|---|
| k=1 | $0.10 \pm 0.02$ | 0.43 | 0.3 | Global object shape | r = 0.78 |
| k=2 | $0.10 \pm 0.02$ | 0.31 | 1.1 | Object boundaries | r = 0.85 |
| k=3 | $0.10 \pm 0.02$ | 0.18 | 2.4 | Fine textures | r = 0.71 |
| k=4 | $0.10 \pm 0.02$ | 0.08 | 3.2 | Noise/artifacts | r = 0.34 |

**Three-Phase Training Dynamics.** Our analysis reveals distinct learning phases with measurable specialization metrics:

Table 14: Three-phase frequency learning progression with quantitative specialization metrics showing evolution from uniform exploration to structured hierarchy.

| Phase | Epochs | Specialization $\sigma$ | Coefficient Variation | Attention Entropy | Stability |
|---|---|---|---|---|---|
| Exploration | 0-40 | 0.02 | 0.12 | $3.41 \pm 0.18$ | $< 30\%$ |
| Specialization | 40-120 | 0.12 | 0.68 | $3.38 \pm 0.12$ | 70% |
| Convergence | 120+ | 0.31 | 0.91 | $3.35 \pm 0.08$ | $> 95\%$ |

This quantitative analysis confirms that different frequency components learn to capture complementary visual patterns, with the strongest correlation (r = 0.85) achieved for object boundary detection at 1.1 Hz.

**Post-Attention vs Pre-Attention Modulation.** We provide comprehensive empirical validation for our design choice:

Table 15: Comprehensive comparison of modulation placement showing superior performance and stability of post-attention design.

| Modulation | ImageNet Top-1 | CIFAR -100 | Gradient $\sigma$ | Convergence | Semantic Preservation | Training Stability |
|---|---|---|---|---|---|---|
| Pre-attention | 82.3% | 82.8% | 0.41 | Epoch 145 | 0.72 | Unstable |
| Post-attention | **84.1%** | **84.26%** | **0.12** | **Epoch 128** | **0.89** | **Stable** |
| Improvement | **+1.8pp** | **+1.46pp** | **-71%** | **-12%** | **+24%** | Qualitative |

**Multi-Head Frequency Specialization.** When allowing head-specific frequency parameters, we observe emergent specialization:

**Head-Specific Parameter Overhead Analysis.** When enabling head-specific Fourier parameters, each attention head learns independent frequency components $\{a_k^{(h)}, \omega_k^{(h)}, \phi_k^{(h)}\}$ and damping coefficient $\gamma^{(h)}$. For a model with $H$ heads and $K$ Fourier components, this increases parameters from $3K + 1$ (shared) to $H \times (3K + 1)$ (head-specific).

The overhead is negligible (0.0003% of model parameters) while providing measurable accuracy improvement (+0.5pp). This demonstrates that frequency specialization across heads is highly parameter-efficient. The per-head frequency distributions show clear differentiation (Table 16): heads naturally divide into low-frequency (global context), mid-frequency (object boundaries), and high-frequency (fine details) groups, with specialization coefficient increasing from 0.31 (shared) to 0.42 (head-specific), indicating stronger differentiation.

**Fourier Components and Damping.** We analyze the impact of each component by selective ablation, as shown in Table 18. Fourier modulation alone provides improvement (+4.43pp) over the RoFormer baseline, while damping alone contributes +2.09pp. When combined, these components achieve a complementary effect, yielding +5.84pp total improvement. Our experiments with varying the number of Fourier components ($K$) show that 4-8 components provides the optimal balance between expressivity and overfitting, with $K = 8$ achieving the best performance (+6.53pp). Similarly, moderate damping ($\gamma$=0.01) yields the best results among the damping coefficients tested.

**Frequency Initialization Strategies.** We also investigate different approaches for initializing the Fourier component frequencies, as shown in Table 19. Logarithmic spacing achieves the best performance (84.62%), providing better coverage across the frequency spectrum compared to linear

Table 16: Multi-head frequency specialization showing automatic division of labor across attention heads with quantitative metrics.

| Configuration | ImageNet Top-1 | Head Group | Freq Range (Hz) | Attention Range | Energy % Energy % | Specialization Timeline |
|---|---|---|---|---|---|---|
| Uniform | 84.1% | All heads | 0.5-1.5 | 45 tokens | 100% | None |
| Head-specific | **84.6%** | Heads 1-2 | 0.2-0.6 | 89 tokens | 35% | Epoch 35 |
| | | Heads 3-4 | 0.6-1.4 | 43 tokens | 40% | Epoch 42 |
| | | Heads 5-6 | 1.4-3.2 | 21 tokens | 25% | Epoch 38 |

Table 17: Head-specific parameter analysis showing modest overhead for improved specialization.

| Configuration | Additional Params | Total Model Params | ImageNet Top-1 | Improvement |
|---|---|---|---|---|
| Shared (baseline) | 13 (4 components) | 24.76M | 83.4% | - |
| Head-specific (6 heads) | 78 (4 comp × 6 heads) | 24.76M (+0.0003M) | 83.9% | +0.5pp |
| *Overhead* | *65 params* | *+0.0003%* | *+0.5pp* | *0.77M params/pp* |

spacing. Random initialization performs worse (83.91%), suggesting that a structured approach to frequency initialization aids optimization. Low-frequency bias initialization shows moderate performance, indicating that while low frequencies are important, a balanced coverage across the spectrum is more effective.

# G COMPUTATIONAL COMPLEXITY ANALYSIS

For completeness, we analyze the computational overhead introduced by the Fourier modulation components in FourierRoFormer. Let $n$ denote the input sequence length, $d$ the feature dimension, and $\kappa$ the number of Fourier components. The computation of the Fourier modulation function requires evaluating $\kappa$ cosine terms for each token pair, computing the modulation, and applying non-linear scaling. Since there are $\mathcal{O}(n^2)$ token pairs in the attention mechanism Vaswani et al. (2017), this results in an overall computational cost of $\mathcal{O}(\kappa n^2)$ operations Rahimi & Recht (2008); Tancik et al. (2020).

**Comprehensive Efficiency Comparison with Spectral Methods.** We provide detailed efficiency analysis comparing FourierRoFormer with spectral transformer baselines:

**Efficiency Metrics Defined:**

- **Efficiency Score** = $\frac{\text{Top-1 Accuracy}}{\log(\text{Params}) \times \sqrt{\text{Training Time}}}$ (higher is better)

- **Parameter Efficiency** = $\frac{\text{Top-1 Accuracy}}{\text{Params (M)}}$ (accuracy per million parameters)

- **Computational Efficiency** = $\frac{\text{Top-1 Accuracy}}{\text{FLOPs (G)}}$ (accuracy per GFLOP)

**Key Findings:** The approach introduces minimal overhead with only 0.04% parameter increase and 0.7% FLOPs increase over RoFormer. It achieves a superior tradeoff with 23% better efficiency score than the best spectral baseline while using 25% fewer parameters. The method provides practical advantage by maintaining standard transformer architecture compatibility unlike spectral methods requiring architectural overhaul.

The additional computational cost of FourierRoFormer compared to standard ViT or RoFormer is minimal, with only 0.01M additional parameters (0.04%) from the learnable Fourier components. During inference, FourierRoFormer processes approximately 215 images/second on our medium model configuration for CIFAR-100, compared to 220 images/second for RoFormer and 218 images/second for standard ViT on identical hardware, demonstrating negligible runtime overhead for improved accuracy gains.

Table 18: Comprehensive ablation study on CIFAR-100 showing complementary benefits of components.

| Configuration | Accuracy (%) | Δ vs RoFormer | Params (M) | GFLOPs |
|---|---|---|---|---|
| RoFormer (baseline) | 78.42 | - | 24.75 | 4.60 |
| + Fourier only | 82.85 | +4.43 | 24.75 | 4.61 |
| + Damping only | 80.51 | +2.09 | 24.75 | 4.60 |
| + Both (Full model) | 84.26 | +5.84 | 24.76 | 4.63 |
| *Fourier Component Variations* | | | | |
| K=2 components | 82.54 | +4.12 | 24.75 | 4.61 |
| K=4 components | 84.26 | +5.84 | 24.76 | 4.63 |
| K=8 components | 84.95 | +6.53 | 24.76 | 4.63 |
| K=16 components | 84.72 | +6.30 | 24.77 | 4.64 |
| *Damping Coefficient Analysis* | | | | |
| $\gamma = 0.001$ | 83.45 | +5.03 | 24.76 | 4.63 |
| $\gamma = 0.01$ | 84.26 | +5.84 | 24.76 | 4.63 |
| $\gamma = 0.05$ | 83.87 | +5.45 | 24.76 | 4.63 |
| $\gamma = 0.1$ | 82.93 | +4.51 | 24.76 | 4.63 |

Table 19: Comparison of frequency initialization strategies on CIFAR-100.

| Strategy | Accuracy (%) | Description |
|---|---|---|
| Linear spacing | 84.26 | Frequencies evenly spaced 0.1-2.0 |
| Logarithmic spacing | **84.62** | Log-spaced frequencies |
| Random initialization | 83.91 | Random frequencies 0.1-2.0 |
| Low-frequency bias | 84.08 | Emphasis on low frequencies |

Table 20: Comprehensive efficiency analysis showing FourierRoFormer achieves optimal accuracy-efficiency tradeoff compared to spectral transformer methods.

| Method | Params (M) | Memory (GB) | Throughput (img/s) | Training Time (h) | FLOPs (G) | Top-1 (%) | Efficiency Score | Parameter Efficiency |
|---|---|---|---|---|---|---|---|---|
| RoFormer-M | 24.75 | 18.0 | 220 | 12.0 | 4.60 | 81.9 | 3.33 | 3.31 |
| GFNet-H-B | 54.0 | 21.5 | 185 | 16.8 | 8.6 | 82.9 | 2.41 | 1.54 |
| WaveViT-B | 33.5 | 19.8 | 195 | 15.2 | 6.8 | 84.8 | 2.98 | 2.53 |
| SpectFormer-H-B | 33.1 | 19.2 | 195 | 14.5 | 6.3 | 85.1 | 3.21 | 2.57 |
| SVT-H-B | 32.8 | 19.5 | 190 | 15.8 | 6.5 | 85.2 | 3.18 | 2.60 |
| FourierRoFormer-M | 24.76 | 18.1 | 215 | 12.3 | 4.63 | 84.1 | **3.91** | **3.40** |
| *Efficiency Advantage vs Best Spectral Baseline (SVT-H-B)* | | | | | | | | |
| Relative Advantage | **-24.5%** | **-7.2%** | **+13.2%** | **-22.2%** | **-28.8%** | **-1.1pp** | **+23%** | **+31%** |